# Fast, Precise Thompson Sampling for Bayesian Optimization

**David Sweet**
Department of Computer Science
Yeshiva University
New York, NY 10174
david.sweet@yu.edu

## Abstract

Thompson sampling (TS) has optimal regret and excellent empirical performance in multi-armed bandit problems. Yet, in Bayesian optimization, TS underperforms popular acquisition functions (e.g., EI, UCB). TS samples arms according to the probability that they are optimal. A recent algorithm, P-Star Sampler (PSS), performs such a sampling via Hit-and-Run. We present an improved version, Stagger Thompson Sampler (STS). STS more precisely locates the maximizer than does TS using less computation time. We demonstrate that STS outperforms TS, PSS, and other acquisition methods in numerical experiments of optimizations of several test functions across a broad range of dimension. Additionally, since PSS was originally presented not as a standalone acquisition method but as an input to a batching algorithm called Minimal Terminal Variance (MTV), we also demonstrate that STS matches PSS performance when used as the input to MTV.

## 1 Introduction

Bayesian optimization (BO) is applied to many experiment- and simulation-based optimization problems in science and engineering [17]. The aim of BO methods is to minimize the number of measurements needed to find a good system configuration. Measurements are taken in a sequence of batches of one or more *arms* – where an arm is one system configuration. System performance is measured for each arm in a batch, then a new batch is produced, performance is measured, and so on.

Thompson sampling (TS) samples arms according to the probability that they maximize system performance [22]. Let's denote an arm as $x_a$, its performance as $f(x_a)$, and the probability that an arm is best, i.e., that $x_a = \mathrm{argmax}_x f(x)$, as $p_*(x)$. TS draws an arm as: $x_a \sim p_*(x)$. TS has optimal or near-optimal regret [1, 3] for multi-armed bandits (MAB) and is also used in BO [9].

Application of TS to BO is not straightforward. Since BO arms are typically continuous – e.g., $x \in [0,1]^d$ – sampling from $p_*(x)$ is non-trivial. The usual approach is to first sample many candidate arms uniformly, $x_i \sim \mathcal{U}([0,1]^d)$, then draw a value, $y_i$, from a model distribution of $f(x)$ at each $x_i$. The $x_i$ that yields the highest-valued draw, i.e., $x_a = \mathrm{argmax}_{x_i} y(x_i)$, is taken as the arm. BO methods usually model $y(x)$ with a Gaussian process, $\mathcal{GP}$ [16], i.e. $y(x) \sim \mathcal{N}(\mu(x), \sigma^2(x))$. While appealing for its simplicity, TS, implemented as described, tends to underperform popular acquisition functions such as Expected Improvement (EI) [10] or Upper Confidence bound (UCB) [5] (see Subsection 3.2).

Workshop on Bayesian Decision-making and Uncertainty, 38th Conference on Neural Information Processing Systems (NeurIPS 2024).

## 2 Stagger Thompson Sampling

The TS arm candidates, being uniform in $[0,1]^d$, are unlikely to fall where $p_*(x)$ has high density. If we imagine that the bulk of $p_*(x)$ lies in a hypercube of side $\varepsilon < 1$, then the probability that a randomly-chosen $x \in [0,1]^d$ falls in the hypercube is just the hypercube volume, $v = \varepsilon^d$. Note that (i) $v$ decreases exponentially with dimension, and (ii) $\varepsilon$, thus $v$, decreases with each additional measurement added to the model of $p_*(x)$ (see figure 5(c) in appendix A). Effect (ii) is the aim of BO – to localize the maximizer. Effect (i) is the "curse of dimensionality". Thus, the number of candidates required to find the bulk of $p_*(x)$ (i) increases exponentially in dimension, and (ii) increases with each additional measurement (albeit in a non-obvious way).

The P-Star Sampler (PSS) [17] is a Hit-and-Run [19] sampler with a Metropolis filter [7]. Our algorithm, Stagger Thompson Sampler (STS), is, also, but differs in several details, discussed below algorithm 1.

---

**Algorithm 1** Stagger Thompson Sampler

1: **if** no measurements yet **then**
2: $\quad \lfloor \quad$ **return** $x_i \sim \mathcal{U}([0,1]^d)$ $\hfill \triangleright$ *Take $p_*(x)$ prior as uniform in $x$*
3: $\tilde{x}_* = \arg\max_x \mu(x)$ $\hfill \triangleright$ *$\mu(x)$ is mean of a given $\mathcal{GP}$*
4: $x_a = \tilde{x}_*$ $\hfill \triangleright$ *Initialize arm*
5: **for all** $m \in 1, \ldots, M$ **do** $\hfill \triangleright$ *Refine arm $M$ times*
6: $\quad \mid \quad x_t = \mathcal{U}([0,1]^d)$ $\hfill \triangleright$ *Perturbation target*
7: $\quad \mid \quad s = e^{-k\mathcal{U}([0,1])}$ $\hfill \triangleright$ *A "stagger" perturbation length*
8: $\quad \mid \quad x_a' = x_a + s(x_t - x_a)$ $\hfill \triangleright$ *Perturbations*
9: $\quad \mid \quad [y, y'] \sim \mathcal{GP}([x_a, x_a'])$ $\hfill \triangleright$ *Joint sample*
10: $\quad \lfloor \quad x_a \leftarrow x_a'$ **if** $y' > y$ $\hfill \triangleright$ *MH acceptance*
11: **return** $x_a$ $\hfill \triangleright$ *A sample from $p_*(x)$*

---

Algorithm 1 modifies vanilla Hit-and-Run in two ways: (i) Instead of initializing randomly, we initialize an arm candidate, $x_a$, at $\tilde{x}_*$. (ii) Instead of perturbing uniformly along some direction – since the scale of $p_*(x)$ is unknown and may be small – we choose the length of the perturbation uniformly in its exponent, i.e. as $\sim e^{-kU}$, a log-uniform random variable. ($k$ is a hyperparameter, which we choose to be $k = \ln 10^{-6}$). Numerical ablation studies in appendix B show that these modifications improve performance in optimization. We refer to the log-uniform perturbation as a "stagger" proposal, following [21]. Besides being empirically effective, a stagger proposal also obviates the need to adapt the scale of the proposal distribution, as is done in PSS (which uses a Gaussian proposal). We see this as a valuable, practical simplification.

Since a log-uniform distribution is a symmetric proposal we expect the Markov chain generated by algorithm 1 to converge to $p_*(x)$. Appendix A provides some numerical support for this.

Perturbations are made along a line from $x_a$ to a target point, $x_t$, which is chosen uniformly inside the bounding box, $[0,1]^d$. This ensures that the final perturbation [a convex combination of points in the (convex) bounding box] will lie inside the bounding box. It also simplifies the implementation somewhat, since boundary detection is unnecessary. PSS performs a bisection search to find the boundary of the box along a randomly-oriented line passing through $x_a$.

We accept a perturbation of $x$, $x'$, with Metropolis acceptance probability $p_{acc} = \min\{1, p_*(x')/p_*(x)\}$. As a coarse (and fast) approximation to $p_{acc}$, we follow PSS and just take a single joint sample from $\mathcal{GP}$ and accept whichever point, $x$ or $x'$, has a larger sample value. Note that this is a Thompson sample from the set $\{x, x'\}$, so we might describe STS as iterated Thompson sampling.

Appendix A offers numerical evidence that (i) samples from STS are nearer the true maximizer than are samples from TS, and (ii) STS produces samples more quickly than standard TS while better approximating $p_*(x)$.

Previous work applying MCMC methods to Thompson sampling include random-walk Metropolis algorithms constrained to a trust region [25] and a sequential Monte Carlo algorithm [2].

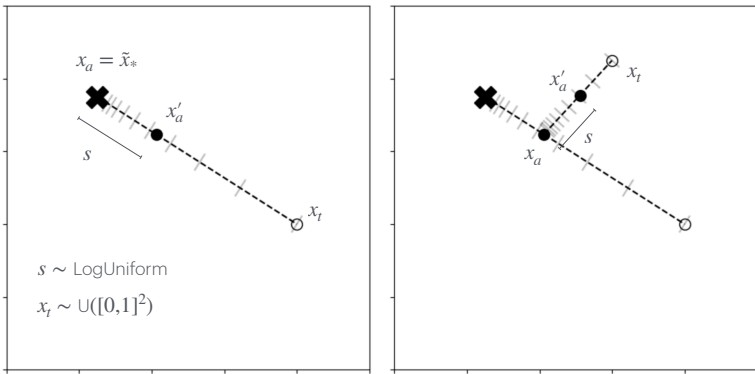

Figure 1: Two iterations of the for loop in algorithm 1. Hash marks indicate the log-uniform (stagger) distribution for $s$. A Thompson sample – a joint sample, $\mathcal{GP}([x_a, x_a'])$ – determines whether $x_a$ updates to $x_a'$.

## 3 Numerical Experiments

To evaluate STS, we optimize various test functions, tracking the maximum measured function value at each round and comparing the values to those found by other methods. We use the term *round* to refer to the generation of one or more arms followed by the measurement of them.

### 3.1 Ackley-200d

To introduce our comparison methodology, we compare STS to a few other optimization methods, in particular to TuRBO, a trust-region-enhanced Thompson sampling method [6]. In [6], the authors optimize the Ackley function in 200 dimensions with 100 arms/round on a restricted subspace of parameters. Figure 2 optimizes the same function on the standard parameter space using STS, TuRBO, and other methods. STS finds higher values of $y$ more quickly than the other methods.

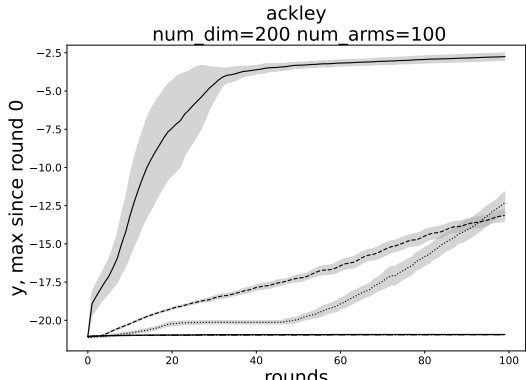

Figure 2: We maximize the Ackley function in 200 dimensions over 100 rounds of 100 arms/round. The error areas are twice the standard error over 10 runs. STS (`sts`) finds higher values more quickly than other optimization methods: `turbo-1` - TuRBO [6] with one trust region. `cma` - CMA-ES [8], an evolution strategy. `random` - Choose arms uniformly randomly (serving as a baseline). [6].

We can summarize each method's performance in Figure 2 with a single number, which we'll call the *score*. At each round, $i$, find the maximum measured values so far for each method, $m$: $y_{i,m}$. Rank these values across $m$ and scale: $r_{i,m} = [\text{rank}(y_{i,m}) - 1]/(M - 1)$, where $M$ is the number of methods. Repeat this for every round, $i$, then average over all $R$ rounds to get the score: $s_m = \sum_i^R r_{i,m}/R$. The scores in figure 2 are $s_{\text{sts}} = 1$, $s_{\text{turbo-1}} = 2/3$, $s_{\text{cma}} = 1/2$, and $s_{\text{random}} = 0$.

Using an normalized score enables us to average over runs on different functions (which, in general, have different scales for $y$). Using a rank-based score prevents a dramatic result, like the one in figure 2, from dominating the average.

In our experiments below we optimize over nine common functions. To add variety to the function set and to avoid an artifact where an optimization method might coincidentally prefer to select points near a function's optimum (e.g., at the center of the parameter space), we randomly distort each function as in [17], repeating the optimization 30 times with different random distortions.

## 3.2 One arm per round

We compare STS to other BO methods in dimensions 3 through 300, all generating one arm per round. For each dimension, each method's score is averaged over all test functions. See Figure 3. STS has the highest score in each dimension, and its advantage appears to increase with dimension. Data from dimensions 1, 10, and 100 (unpublished for space) follow the same pattern.

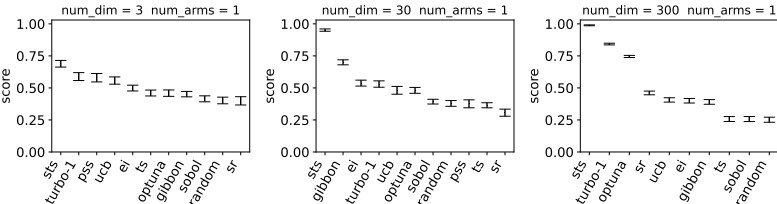

Figure 3: We optimize for $\max(30, \texttt{num\_dim})$ rounds with $\texttt{num\_arms}$ / round over the functions ackley, dixonprice, griewank, levy, michalewicz, rastrigin, rosenbrock, sphere, and stybtang [20] with random distortions (see section 3.1). Error bars are two standard errors over all functions and 30 runs/function. Figure represents a total of 874,800 function evaluations. (We were not able to run `pss` for num_dim=300 due to long computation times.

The various optimization methods are: `sts` - Stagger Thompson Sampling (Algorithm 1), `random` - Uniformly-random arm, `sobol` - Uniform, space-filling arms [18, Chapter 5], `sr` - Simple Regret, $\mu(x)$, `ts` - Thompson Sampling [9], `ucb` - Upper Confidence Bound [5, 24], `ei` - Expected Improvement [12, 24], `gibbon` - GIBBON, an entropy-based method [13], and `optuna` - an open-source optimizer [15] employing a tree-structured Parzen estimator [23]. For the methods `ei`, `lei`, `ucb`, `sr`, `gibbon`, and `turbo-1`, we initialize by taking a Sobol' sample for the first round's arm. `sts` and `ts` do not require initialization.

STS makes no explicit accommodations for higher-dimensional problems yet performs well on them. Of the methods evaluated, only `turbo-1` specifically targets higher-dimensional problems [6], so it may be valuable for future work to compare STS to other methods specifically designed for such problems. (See references to methods in [6].)

## 3.3 Multiple arms per round

Thompson sampling can be extended to batches of more than one arm simply by taking multiple samples from $p_*(x)$, e.g., by running Algorithm 1 multiple times. However, this approach can be inefficient [14] because some samples – since they are independently generated – may lie very near each other and, thus, provide less information about $f(x)$ than if they were to lie farther apart. This problem, that of generating effective batches of arms, is not unique to TS but exists for all approaches to acquisition, and there are various methods for dealing with it [14] [24] [13].

One method, Minimal Terminal Variance (MTV) [17], minimizes the post-measurement, average variance of the GP, weighted by $p_*(x)$:

$$MTV(x_a) = \int dx \, p_*(x)\sigma^2(x|x_a) \tag{1}$$

approximated by $\sum_i \sigma^2(x_i|x_a)$, where $x_i$ are drawn from $x_i \sim p_*(x)$ with P-Star Sampler (PSS). MTV is interesting, in part, because it can design experiments both when prior measurements are available and *ab initio* (e.g., at initialization time). It not only outperforms acquisition functions

(like EI or UCB) but the same formulation also outperforms common initialization methods, such as Sobol' sampling [17].

We modify MTV to draw $x_i \sim p_*(x)$ using STS instead of PSS. Note, also, that the arms, $x_a$, that minimize $MTV(x_a)$ are not drawn from the set $x_i$ but are chosen by a continuous minimization algorithm (specifically, `scipy.minimize`, as implemented in [11]), such that $x_a = \operatorname{argmin}_{x'_a} \sum_i \sigma^2(x_i|x'_a)$.

Figure 4 compares MTV, with P-Star Sampler replaced by STS (`mtv+sts`), to other methods using various dimensions (`num_dim`) and batch sizes (`num_arms`): `mtv` - MTV, as in [17], `lei` - q-Log EI, an improved EI [4], and `dpp` - DPP-TS [14], a diversified-batching TS. For the methods `ei`, `ucb`, `sr`, `gibbon`, and `dpp`, we initialize by taking Sobol' samples for the first round. `sts`, `mtv`, and `mtv+sts` do not require initialization.

Figure 4 roughly reproduces figure 3 of [17], adding more methods and extending to higher dimensions. Additionally, we include pure PSS and STS sampling, where arms are simply independent draws from $p_*(x)$, to highlight the positive impact of MTV on batch design.

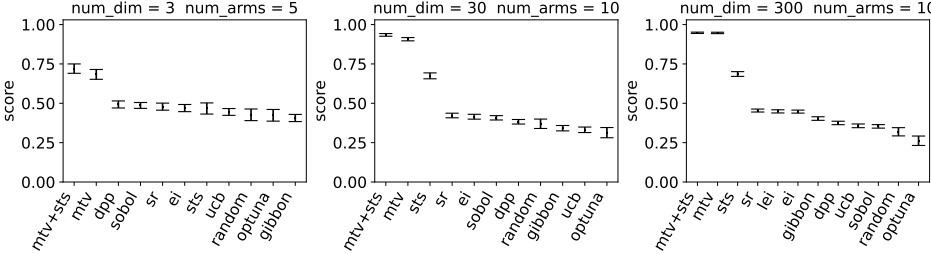

Figure 4: Optimizations with 3 multi-arm rounds on nine test functions. MTV+STS (`mtv+sts`) outperforms all other methods across a range of dimensions. The figure consists of $1.2 \cdot 10^6$ function evaluations. Calculations (not shown) for 1, 10, and 100 dimensions show similar results.

When MTV's input samples come from STS (`mtv+sts`), performance is similar to the original MTV (`mtv`).

## 4    Conclusion

We presented Stagger Thompson Sampler, which is novel in several ways:

- It outperforms not only the standard approach to Thompson sampling but, also, popular acquisition functions, a trust region Thompson sampling method, and an evolution strategy.
- It is simpler and more effective at acquisition than an earlier sampler (P-Star Sampler).
- It works on high-dimensional problems without modification.

Additionally, the combination MTV+STS is unique in that it applies to so broad a range of optimization problems: It solves problems with zero or more pre-existing measurements, with one or more arms/batch, and in dimensions ranging from low to high.

### Acknowledgments and Disclosure of Funding

This work was carried out in affiliation with Yeshiva University. The author is additionally affiliated with DRW Holdings, LLC. This work was supported, in part, by a grant from Modal.

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

# A  Speed, Precision, and Thompson Sampling

In this appendix we provide numerical support for the claims of the paper title.

Figure 5 compares STS to PSS and standard Thompson sampling using various numbers of candidate arms (100, 3000, and 10000). For each Thompson sampling method we maximized a sphere function

$$f(x) = -(x - 0.65\mathbf{1})^2$$

in the domain $x \in [0,1]^5$ where $\mathbf{1}$ is the vector of all 1's. At each round of the optimization we drew one arm, refit the GP, then drew 64 Thompson samples, $x_i$, solely for use in calculating summary statistics (i.e., not for use in the optimization). We have also included a uniformly-random sampler (`sobol`, not a Thompson sampler) for comparison. The samples were generated by a Sobol' quasi-random sampler [18, Chapter 5].

**Precision**: Subfigure (a) shows $rmse = \sum_i (x_i - 0.65)^2/64$, describing how near the Thompson samples are to the true optimum, $x = 0.65\mathbf{1}$. Subfigures (b) and (c) decompose the RMSE into $bias = \sum_i (x_i - 0.65)/64$ and $scale = \left(\Pi_d s_d\right)^{1/5}$, where $s_d$ is the standard deviation of the $d^{\text{th}}$ dimension of the samples $x_i$. We note that the Thompson samples, $x_i$, get closer to the maximizer as (i) we increase the number of candidates in standard TS, (ii) the optimization progresses and more measurements are included in the GP, (iii) we switch from standard TS to PSS to STS. We note, also, that all Thompson samplers produce similarly unbiased samples, and that the improvement in RMSE comes from greater precision, i.e., reduced scale of the distribution of $x_i$.

**Thompson Sampling**: Next we support our claim that $x_i \sim p_*(x)$ by calculating $p_{\max,i}$, an estimate of the probability that sample $x_i$ is the maximizer over the 64 samples. To calculate $p_{\max,i}$ we take a joint sample $y_i \sim \mathcal{GP}(x_i)$ over the 64 $x_i$ and record which $x_i$ yields the largest $y_i$. We repeat this 1024 times and set $p_{\max,i} = $ [count of times $x_i$ is the max]/1024. The subfigure `std(p_max)` shows the standard deviation of $p_{\max,i}$ over $i$. If all 64 samples $x_i$ were Thompson samples then we'd expect $p_{\max,i} = 1/64$ and $\text{std}(p_{\max}) = 0$. We see that $\text{std}(p_{\max})$ stays closer to zero for both PSS and STS, while the values for standard TS grow as the optimization progresses, similar to the uniformly-random sampler (`sobol`). [While low $\text{std}(p_{\max})$ is a necessary condition to claim that $x_i \sim p_*(x)$, it is not sufficient. For instance, there may be regions of $[0,1]^d$ where $p_*(x) > 0$ but no $x_i$ appear.]

**Speed**: The running time (in seconds, subfigure (e)) is smaller for STS than for PSS or for standard TS with 10,000 candidates. Note that the y-axis has a logarithmic scale to show the separation between curves, although `duration` is linear in `round`. Subfigure (f) verifies that all methods optimize the sphere function. We configured PSS as in [17]. (It is unclear whether the number of iterations used by the Hit-and-Run sampler was optimal or whether PSS could have been faster or slower if this number were tuned. In the next section, section B, we tune the number of iterations used by STS's Hit-and-Run and show that the value we used in the paper is optimal.)

Point (i), above, suggests that given enough candidates, TS might achieve the same small scale that STS does, although this would increase the running time of TS, and it is already much larger that of STS even at only 10,000 candidates.

# B  Ablation studies

Stagger Thompson Sampling (STS), algorithm 1, modifies a Hit-and-Run sampler in two ways: (i) Instead of initializing $x_a$ randomly, it uses a guess at the maximizer, $\tilde{x}_* = \text{argmax}_x \mu(x)$, and (ii) perturbation distances are drawn from a log-uniform distribution rather than uniformly.

Figure 6 compares various ablations of STS:

- `sts-ui` initializes $x_a$ uniformly-randomly
- `sts-m` initializes $x_a$ with the $x$ having the highest previously-measured $y$ value
- `sts-t` initializes $x_a$ to a Thompson sample from the $\mathcal{GP}$ at previously-measured $x$ values
- `sts-ns` replaces the stagger (log-uniform) perturbation with a uniform one

PSS, standard TS, and random arm selection are included for scale. The figure shows that changing any of the features itemized above can reduce performance of STS.

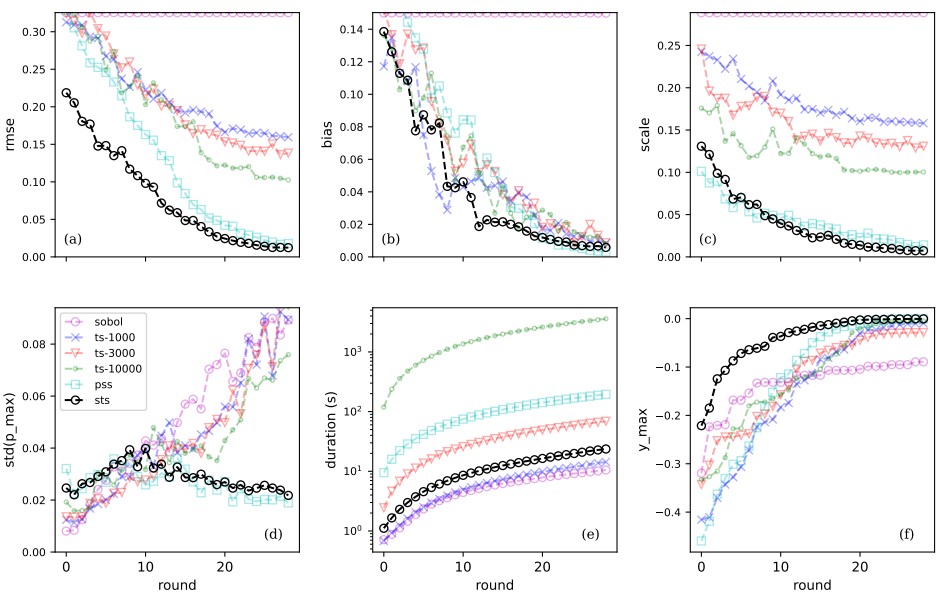

Figure 5: Comparison of STS to PSS and standard TS with varying numbers of candidates (1000, 3000, and 10,000). See appendix A for discussion. The optimizer, `sobol`, which proposes arms uniformly randomly, is included as a baseline.

Figure 7 sweeps values of a parameter, $M$, to STS, the number of iterations of the Hit-and-Run walk. See algorithm 1 for details. The figure shows that performance asymptotes around $M = 30$, which is the value used throughout the paper.

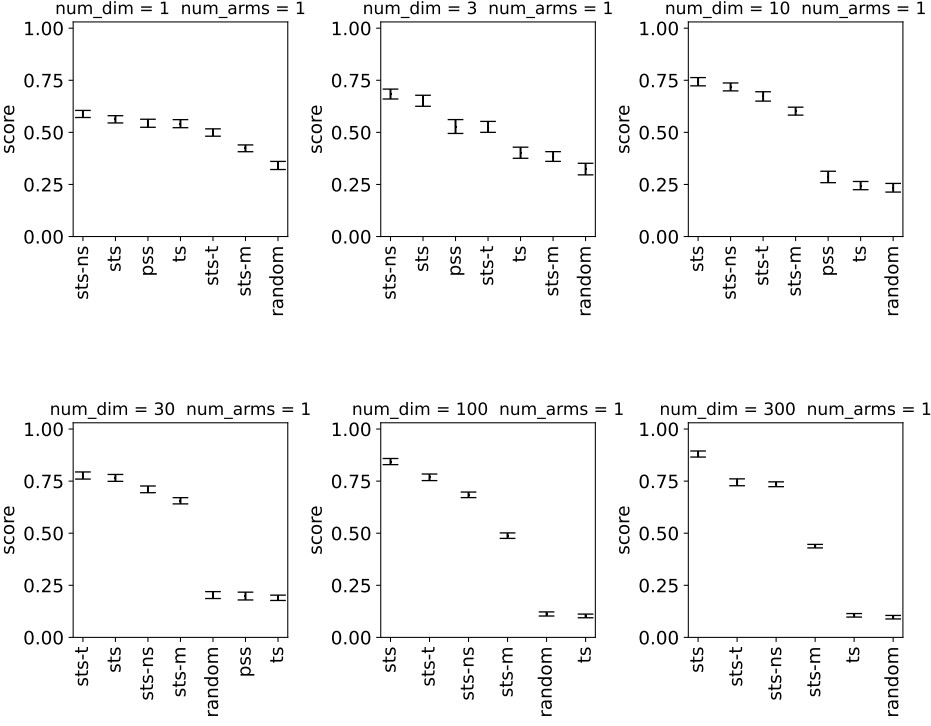

Figure 6: Ablations. `sts`, as presented in algorithm 1, performs as well as or better than any of the ablated version evaluated here. See text for descriptions of ablations..

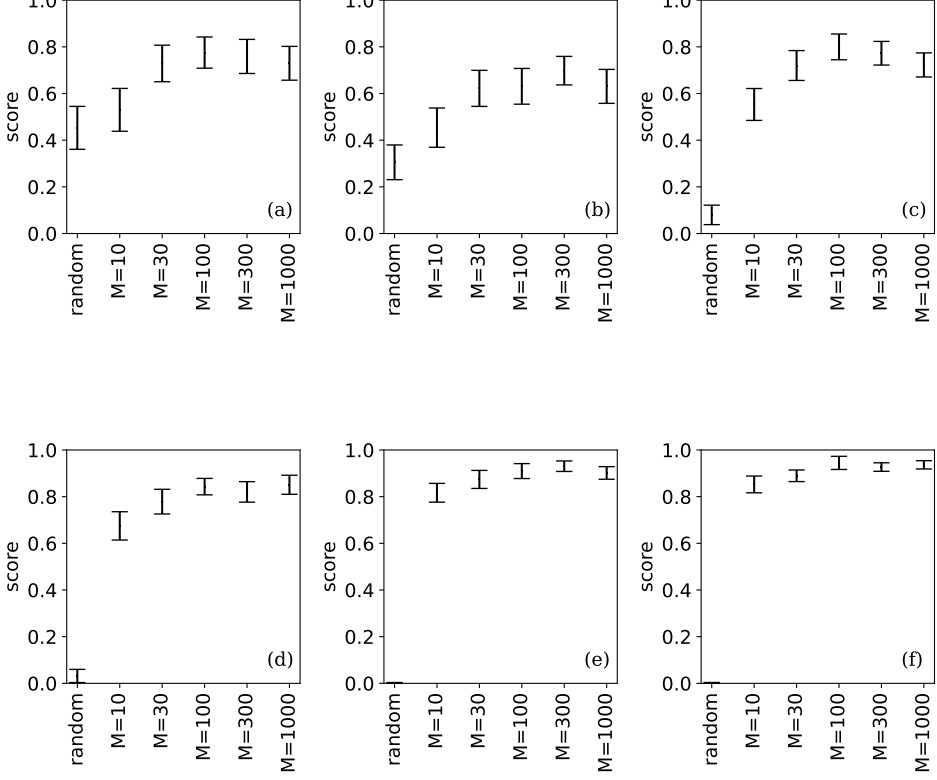

Figure 7: Performance stabilizes around $M = 30$. Larger values of $M$ would increase running time for no meaningful benefit.

