# OpenReview forum: "Fast, Precise Thompson Sampling for Bayesian Optimization"
_NeurIPS.cc/2024/Workshop/BDU — NeurIPS BDU Workshop 2024 Poster_

### Official Review · Reviewer_N6Ju · 2024-09-19
**Review of the submission**

**Rating:** 4
**Confidence:** 5

**Review:**

The article presents an MCMC-inspired method for sampling from the Thompson sampling distribution in Gaussian process bandits, which is also a credible (if slightly misguided) approach to Bayesian optimisation - in pure Bayesian optimisation we should not care about incurring poor performance on the way through, whereas bandits and TS do need to care about just how bad some arms might be. Nevertheless, I am reviewing purely whether this is a good article on how to perform Thompson sampling in GPs.

The approach builds on the p* sampler of Ren and Sweet. What is proposed is not daft, and seems to work well in the presented experiments. However I have significant misgivings about the clarity of the paper, and think it is not ready for presentation.

My three key concerns are:
1. The algorithm is presented in a reproducible way, but there is no analysis of what the stationary distribution of this Markov chain is, and therefore no justification for why it might be sampling from approximately the Thompson sampling distribution.
2. The interfacing of MTV with the approach in section 3.3 is not at all well explained. Is it the case that too many Thompson samples are produced then MTV thins these down?
3. It is claimed that the new method is significantly computationally faster than the p* sampler on which it builds. But both are simply MCMC schemes with an arbitrary run length. I cannot understand how a reasonable comparison of computational effort can be made, and certainly no attempt is made to do so.

More minor comments, to assist a future revision:
- Use of "ex. " instead of "for example" or "eg" is non-standard and quite confusing
- In the abstract, the use of "arm dimension" is a little jarring for some reason. Perhaps "action space dimension"?
- In l32, v is the volume of a fixed sized box. It doesn't change with data. So this statement is incorrect.
- In Alg 1, what is p_max?
- It's unfortunate that the key prior work cannot be implemented in the first example (l59). It would be much better to start with an example where the prior work can be implemented then to move on.

The topic is kind of interesting for the workshop, and it's possible that a poster would be interesting. But I suspect there will be better submissions.

---

### Official Review · Reviewer_J8v1 · 2024-09-28
**Review of Submission 6**

**Rating:** 6
**Confidence:** 3

**Review:**

**Summary:** This work presents a novel approach for fast and exact Thompson sampling in Gaussian process-based Bayesian optimization. The authors introduce an MCMC scheme that enables efficient sampling of points based on their probability of being the global optimum. The proposed method shows significant performance gains in various numerical experiments.

**Strengths:** The method is straightforward to implement, and the empirical results are impressive, demonstrating notable improvements over existing approaches.

**Weaknesses:** My primary concern is the lack of clarity regarding the "exactness" of the proposed sampling scheme. It is not immediately clear that the introduced MCMC approach samples from the posterior distribution over the optimum. A detailed discussion addressing this aspect is necessary.

---

### Decision · Program_Chairs · 2024-10-09

Accept (Poster)